# Optimized molecule detection in localization microscopy with selected false positive probability

**Miroslav Hekrdla** [1] ✉, **David Roesel** [1], **Niklas Hansen** [1,2], **Soumya Frederick**[1,2], **Khalilullah Umar**[1,3] **& Vladimíra Petráková** [1] ✉

Single-molecule localization microscopy (SMLM) allows imaging beyond the diffraction limit. Detection of molecules is a crucial initial step in SMLM. False positive detections, which are not quantitatively controlled in current methods, are a source of artifacts that affect the entire SMLM analysis pipeline. Furthermore, current methods lack standardization, which hinders reproducibility. Here, we present an optimized molecule detection method which combines probabilistic thresholding with theoretically optimal filtering. The probabilistic thresholding enables control over false positive detections while optimal filtering minimizes false negatives. A theoretically optimal Poisson matched filter is used as a performance benchmark to evaluate existing filtering methods. Overall, our approach allows the detection of molecules in a robust, single-parameter and user-unbiased manner. This will minimize artifacts and enable data reproducibility in SMLM.

Single-molecule localization microscopy (SMLM) encompasses powerful techniques that surpass the diffraction limit of light[1,2]. These advancements have lead to breakthroughs in fields such as cellular or molecular biology. The process of molecule detection in microscopy images is integral to SMLM methodologies. It is an essential first step for extracting quantitative information from microscopy images and forms a basis for all subsequent analysis[3]. The detection algorithm aims to identify individual molecules and determine their pixel-based locations, effectively separating molecular signals from background noise. However, the current state of molecule detection in SMLM faces several challenges. The detection process involves numerous methods, each with multiple settings that users manually adjust. This complexity makes the process difficult to execute, reproduce, and interpret without errors or artifacts[4]. Furthermore, the selection of the detection method in localization software is often ad hoc lacking a systematic approach. The detection methods implemented across various localization software differs[5,6]. This restricts users to potentially sub-optimal performance and limits the reproducibility of data analysis.

The variety and complexity of current detection methods are substantial. Briefly, these methods include linear filters such as mean filters, Gaussian filters[5,7], and more complex compositions like Laplacian of Gaussian[8] and difference of Gaussians. Nonlinear filters, such as median filters, and advanced noise suppression techniques like bilateral filtering and nonlocal means, are also employed[9]. Morphological operations, including top-hat filters and h-dome transforms, play a role in some approaches[10]. Thresholding techniques range from simple intensity-based thresholds to more sophisticated methods like Otsu's threshold[5,6,11]. Additionally, machine learning approaches, including traditional segmentation methods and emerging deep learning techniques, are becoming increasingly relevant in this field[7,10,12–15]. For details see our overview of single-molecule detection methods in Supplementary Note 1.

A significant issue in the current framework for molecule detection is the lack of quantitative control over detection performance, particularly in terms of false positive and false negative detections. Users typically set parameters based on experience and make subjective judgments about detection accuracy, leading to inconsistent

[1]J. Heyrovský Institute of Physical Chemistry, Czech Academy of Sciences, Prague, Czechia. [2]Department of Physical Chemistry, University of Chemistry and Technology, Prague, Czechia. [3]Faculty of Biomedical Engineering, Czech Technical University in Prague, Kladno, Czechia. ✉e-mail: miroslav.hekrdla@jh-inst.cas.cz; vladimira.petrakova@jh-inst.cas.cz

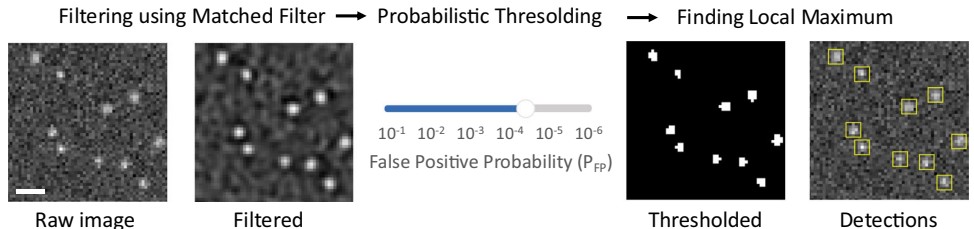

**Fig. 1 | Schematics of the detection.** Proposed molecule detection algorithm using matched filter and adaptive probabilistic thresholding where the user sets the false positive probability $P_{FP}$ to adjust the detection threshold. Scale bar is 1 μm.

results. In SMLM, the negative impact of false positive detections is particularly problematic. False positives can form non-existent artificial structures in the super-resolved image and propagate errors throughout the SMLM processing pipeline, affecting post-processing and quantitative analysis. The variability in detection errors is evident in the outcomes of the SMLM challenge, where different algorithms produced significantly different false positive detections over the same dataset (Supplementary Fig. 9). This variability underscores the need for a more standardized and controlled approach to molecule detection in SMLM. An uncontrolled level of false positive detections acts as noisy data and negatively affects the entire subsequent processing chain (from post-processing steps like drift correction and merging, to quantitative analysis such as clustering, molecule count, quality assessment, particle fusion, and tracking) in unpredictable ways.

To address these challenges, we have developed a user-unbiased detection algorithm that is theoretically optimal (with the highest detection probability), simplifies the process for the user, and quantifies its performance. It is a combination of probabilistic thresholding and optimal filtering. Our approach frames molecule detection within the context of signal detection theory[16], and derives a detection method with the highest theoretical performance that enables control over the level of false positive detections. We combine it with a theoretically optimal Poisson Matched Filter (PMF) using it as performance benchmark to systematically evaluate alternative methods. The benchmark revealed Matched Filter (MF) to be the method of choice. This enables users to ensure optimal and reproducible molecule detection simply by setting a desired false positive probability. Notably, the presented method is readily generalizable to three-dimensional imaging. This approach is well established in astronomy[17] and radar[18], and we now bring its benefits to SMLM. We present an implementation of the method that is fast, robust, and uses standard Python libraries (implementation details are given in Supplementary Note 2).

## Results

### Algorithm description
Our algorithm is comprised of two main steps: filtering and probabilistic thresholding. In this section, we introduce the Poisson-matched filter to set a performance benchmark for other filtering methods. We then show how to apply the probabilistic thresholding for molecule detection, allowing the user to control molecule detection by selecting the probability of false positive detection. Last, we add background adaptation to our method, where the threshold level is calculated from the local background level. Based on the detection performance and computational efficiency, we propose optimized molecule detection that consists of a matched filter and probabilistic thresholding visualized in Fig. 1.

### Optimal filtering
We start by deriving the Poisson Matched Filter (PMF), a theoretically optimal detection filter (with the highest detection probability) for the Poisson noise model, which is well-suited for shot noise-limited

conditions prevalent in microscopy. PMF has already been proposed in X-ray astronomy[19] and optical communication systems[20] but has not appeared in the context of fluorescence microscopy. The derivation uses techniques from the signal detection theory[16,21], particularly the Generalized Neyman-Pearson's Likelihood Ratio Test (GLRT). PMF offers theoretically optimal performance, which means achieving the maximum probability of detection $P_D$ while keeping the probability of false positive $P_{FP}$ equal to a selected level. By solving the GLRT for a Point Spread Function (PSF) generation model in a Poisson noise (see "Methods" for details), we receive weights representing the PMF filter kernel

$$w_{mn}(a,b,\sigma) = \ln\left(1 + \frac{a}{b} s_{mn}(\sigma)\right). \tag{1}$$

The shape of the PMF filter depends on the PSF shape $s_{mn}(\sigma)$, its width $\sigma$, and the molecule contrast, defined here as the signal-to-background ratio $a/b$. We used PMF as a benchmark to systematically assess the performance of other filtering methods used in SMLM. Interestingly, the PMF's shape closely approximates that of other filters across different signal-to-background ratios: matched filter for low contrast $a/b$, and B-spline wavelets for medium to high contrast $a/b$, depending on B-spline parameters (see Fig. 2), which places these filters in a common theoretical framework. The width of the optimal PMF filter increases with increasing ratio $a/b$. This is in agreement with the conclusions of the related work[22], which recommends (based on numerical simulations) to use a Gaussian filter with a larger standard deviation than the one of the PSF.

Because molecules with medium to high contrast are generally detected with high accuracy regardless of the applied filter, it is central to optimize detection performance for molecules with low contrast. Therefore, the matched filter emerges as the preferable option not only because of its optimal performance but also because of its simplicity. Unlike PMF, which requires three parameter estimates ($a, b, \sigma$), the matched filter requires only a single parameter ($\sigma$). We explain the relation of the PMF and matched filter for low contrast molecules in Supplementary Note 3.

In specific cases, the Poisson noise model might not be a suitable approximation for microscopy data, for example when the shot noise is combined with significant read-out noise as in the CCD camera. Then, the model needs to be extended to the Poisson–Gaussian noise[23]. We performed GLRT in Poisson-Gaussian noise showing that it does not lead to a compact formula (Supplementary Note 4). However, if we prioritize the detection of the weak signal with a low signal-to-background ratio $a/b$, the GLRT is equivalent to the matched filter. It demonstrates that the matched filter remains a preferable solution even under the mixed Poisson-Gaussian noise conditions. Another case generalizing detection in the Poisson noise model is the case of the sCMOS camera, where each pixel effectively has a known, different variance of the noise. In Supplementary Note 5, we show that the optimal detection filter is an inversely weighted matched filter, sometimes also referred to as a whitened matched filter.

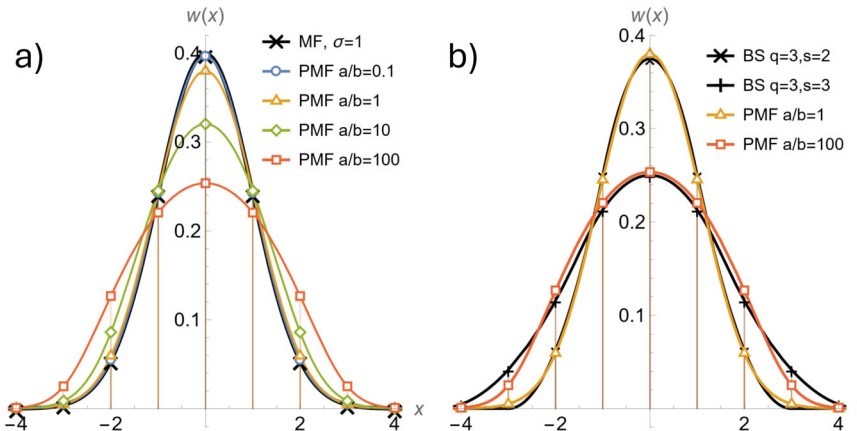

**Fig. 2 | Comparison of the shape of the Poisson Matched Filter (PMF) with other relevant filters. a** PMF for different signal-to-background ratios $a/b$ shows that the shape is practically identical to that of a Matched Filter (MF) for $a/b = 0.1$. **b** Comparison of B-Spline wavelets (BS) and PMF showing clear similarity for mid and high $a/b$. Markers indicate the values used to create the filtering mask.

## Probabilistic thresholding

In the second step, we implement a probabilistic detection threshold to enable quantitative control over false positive detection errors. This method is well-established in the fields of radar[18], astronomy[9], and medicine[24] and has been previously recommended for microscopy and particle tracking in refs. [25–27]. In our methodology, detection performance is optimized by aligning the algorithm's sensitivity with a user-defined false positive probability, effectively calibrating the detection threshold. In practice, this means the user only needs to input the desired false positive probability, which serves as not only the sole parameter for setting the detection threshold but also a detection error estimate.

To use the probability of false positive detection as a parameter that sets the detection threshold, we take advantage of the knowledge of the analytical formula of $P_{FP}$, where $P_{FP}$ depends on the mean background level and on the threshold $\tau$. As the background level can be easily estimated from the images, the only parameter controlling the probability $P_{FP}$ is the decision threshold $\tau$. By inverting the equation, we receive a formula for threshold as a function of $P_{FP}$

$$\tau = \sqrt{\hat{b} \sum_{m,n=-L}^{L} w_{mn}^2 Q^{-1}(P_{FP})} + \hat{b} \sum_{m,n=-L}^{L} w_{mn}, \quad (2)$$

where $\hat{b}$ denotes the mean background estimate, $w_{mn}$ is a filter kernel of the size $(2L+1) \times (2L+1)$ and $Q^{-1}$ is the inverse complementary cumulative distribution function of the standard normal distribution. We refer to threshold $\tau$ as the probabilistic threshold. Probabilistic thresholding can be used in combination with any linear filter common in SMLM. Probabilistic thresholding for a special case of low input signal in EMCCD camera is discussed in Supplementary Note 6.

## Adaptability of thresholding to local background

In typical SMLM images, the background may vary across the field of view both spatially and temporally, changing from frame to frame. The estimation of the background is necessary to set the probabilistic threshold. Any existing background estimation methods can be used for this purpose. From local arithmetic averaging in space, median filtering over time[28], to more complex methods utilizing precise knowledge of the background structure[29], or estimation based on powerful deep neural networks[30]. It is reasonable to estimate the background from values in both space and time if they can be considered as locally constant in this neighborhood generally corresponding to 3D space-time averaging. To complement existing

methods, we propose an averaging method implemented using filtering (Fig. 3). Background estimation is performed from local values, but the nearest neighborhood (approximately the size of PSF) containing pixels biased by the presence of the signal is omitted. The resulting filter kernel is shaped like a 2D or 3D doughnut (see Fig. 3d). This method is widely used in radar[18] (more details in Supplementary Note 7).

We found that the thresholding using adaptive background estimation surpasses considered methods that are robust to background variability or methods that eliminate the background (see Fig. 4). Our results indicate that the widely used practice of background removal might not be theoretically justified and might become unnecessary when using background-adaptive thresholding, pointing to an interesting problem for further studies.

## Generalization for molecule detection in 3D

The presented methodology of molecule detection is general and allows application in 3D. The specific form of implementation depends on the assumed model of signal generation. We distinguish three basic models to outline the detection algorithms. The first one (called here multiplane) is for the case of the full z-stack in all slices of potential molecule z-axis position. An example can be the multifocus microscopy method[31]. The second one (which we call z-parameterized PSF) uses PSF engineering methods that encode the value of the z-axis into a PSF shape. Representatives are for example the astigmatism, double helix, and tetrapod methods. The last considered method is called z-projection detection, where we do not distinguish molecules in the z-axis and the detection is performed on data where the z-component of the data has been marginalized (averaged out). An example is the biplane method[32]. For the overview of the 3D SMLM methods, we rely on references[33,34].

**Detection in multiplane.** Similarly to the 2D case, we seek the maximum of the test statistic (i.e., the filtered signal) over all possible integer shifts of the molecule in all dimensions. In 3D, we search in three dimensions in the received signal, which is no longer a 2D image but a 3D datacube. Using the Poisson noise model and the GLRT test we obtain the optimal Poisson-matched filter (see Supplementary Note 14 for details).

$$w_{lmn} = \ln\left(1 + \frac{a}{b} s_{lmn}\right), \quad (3)$$

where $s_{lmn}$ denotes the 3D PSF. Any analytical model or measured function can be used as $s_{lmn}$.

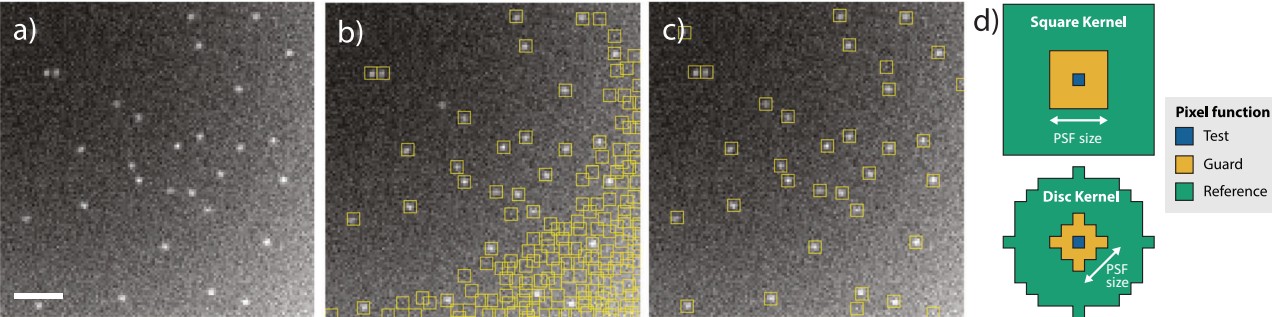

**Fig. 3 | Molecule detection in images with varying background using adaptive thresholding. a** An example of a simulated image of a varied background with 30 molecules placed across the area. Scale bar is 2 μm. **b** Detection artifacts displaying an incorrect number of false positive detections when the threshold is not adaptive to varying background and **c** A solution to this problem by background adaptive thresholding. $P_{FP}$ is set to $10^{-4}$ in both images. **d** Kernels displaying reference pixels (yellow) used for local background estimate and guard pixels (blue) preventing errors due signal present.

**Detection with z-parameterized PSF.** For detection with parameterized PSF, we only have 2D measurements in one axial axis in which we search for the corresponding slice of the parameterized PSF. For this task, we can also use a GLRT detection test where one of the search parameters is the z-coordinate of the molecule, where $w_{mn}(l)$ is the $l$-th slice of the parametrized detection filter. For the Poisson noise model, the detection filter is

$$w_{mn}(l) = \ln\left(1 + \frac{a}{b} s_{mn}(l)\right). \tag{4}$$

Symbol $s_{mn}(l)$ denotes the $l$-th slice of parametrized PSF model.

**Detection in z-projection.** In the z-projection detector, the detection is performed over 3D data where the z-axis component has been averaged out to obtain a 2D model of the signal that is equivalent to the model described in the manuscript, except that now instead of using a 2D PSF function, we use a z-axis averaged 3D PSF

$$\tilde{s}_{mn} = \frac{1}{N_z} \sum_l s_{lmn}. \tag{5}$$

As a proof of concept for the application of detection theory in 3D SMLM, we show an implementation of the 3D detection for the case of multiplane imaging, including a basic performance evaluation (Supplementary Note 8). A comparison of all possible detection filters for all forms of 3D detection is beyond the scope of this work. Here, we demonstrate the general functionality of the method.

**Evaluation of performance and comparison to other methods**
We evaluated our detection algorithm by comparing its performance against several commonly used detection methods in SMLM that differ only in the type of detection filter and its parameters. Table 1 lists these filters alongside the SMLM software in which they are implemented. Definitions of these methods are provided later in "Reference methods".

To assess the performance, we have selected Receiver Operating Characteristic (ROC) curves[35] as a primary performance metric. ROC curves illustrate the relationship between the detection probability $P_D$ on the probability of false positive detection $P_{FP}$. The higher the $P_D$ for given $P_{FP}$, the better the algorithm's performance. While related works often use F1/Dice measures or Jaccard index[5,6,22], we chose ROC evaluation due to the asymmetric impact of false positives and false negatives in molecule detection, where false positives have more severe consequences. This approach is consistent with recent community recommendations on metric selection in image processing tasks[36].

To quantitatively evaluate the performance of detection methods, it is necessary to use datasets with annotated ground truth detections[5,6]. We evaluated detection methods across five different datasets: simulated data with uniform background, simulated data with varying background, two datasets created for the SMLM challenges, and experimental data of static gold beads.

First, we tested the methods using data with constant background, each simulated as a single randomly placed fluorescent molecule on a uniform background with Poisson noise. An example of a generated image is shown in Supplementary Fig. 10. The highest detection probability $P_D$ for any probability of false positive $P_{FP}$ was given by PMF, supporting the theory. The matched filter performed similarly well. The net-gradient method performed the worst, followed by Difference of Gaussians (DoG) and Difference of Arithmetic means (DoA) filters as depicted in Fig. 4a). For illustration, we also show the performance of the methods on this type of data using Jaccard metric in Supplementary Fig. 11.

Second, we have compared the performance on data with varying background and Poisson Noise. We modeled a parabolic type of background that corresponds to imaging conditions of a non-uniform illumination similar to the effect of vignetting[37] as illustrated in Fig. 3. Results are found in Fig. 4b), showing PMF and matched filter as the best performers.

As the next step, we took advantage of the datasets created for the SMLM challenge[5,6], and compared the performance of various filters on the noisy Bundled Tubes Long Sequence (BT-LS) dataset (Fig. 4c) and microtubules, low SNR, low density (MT0_N2_LD) dataset (Supplementary Fig. 12). Results support the theoretical predictions and confirm PMF and MF as the highest performers.

To demonstrate the practical use of the detection algorithm on experimental data, we additionally tested the performance on images of DNA origami-bound gold beads. They are easy to detect under standard visualization conditions due to their high brightness, effectively serving as a reference to annotate ground truth detections. To ensure the homogenous distribution of nanoparticles and reduce agglomeration during the deposition, we used gold nanoparticles that were attached to DNA origami with biotinylated sites and deposited them on BSA-biotin-treated glass coverslips by using neutravidin as a linker. We visualized the samples in two modes. One under standard visualization conditions that result in bright diffraction-limited spots on a low background that served to annotate the ground truth reference. In the second mode, we changed the imaging conditions to make the beads appear less distinct, mimicking how low-contrast molecules might appear in other experiments and used it to test the performance of the detection methods (Fig. 5a). Resulting ROC curves show that PMF and matched filter outperforms others on experimental data (Fig. 5b) as well.

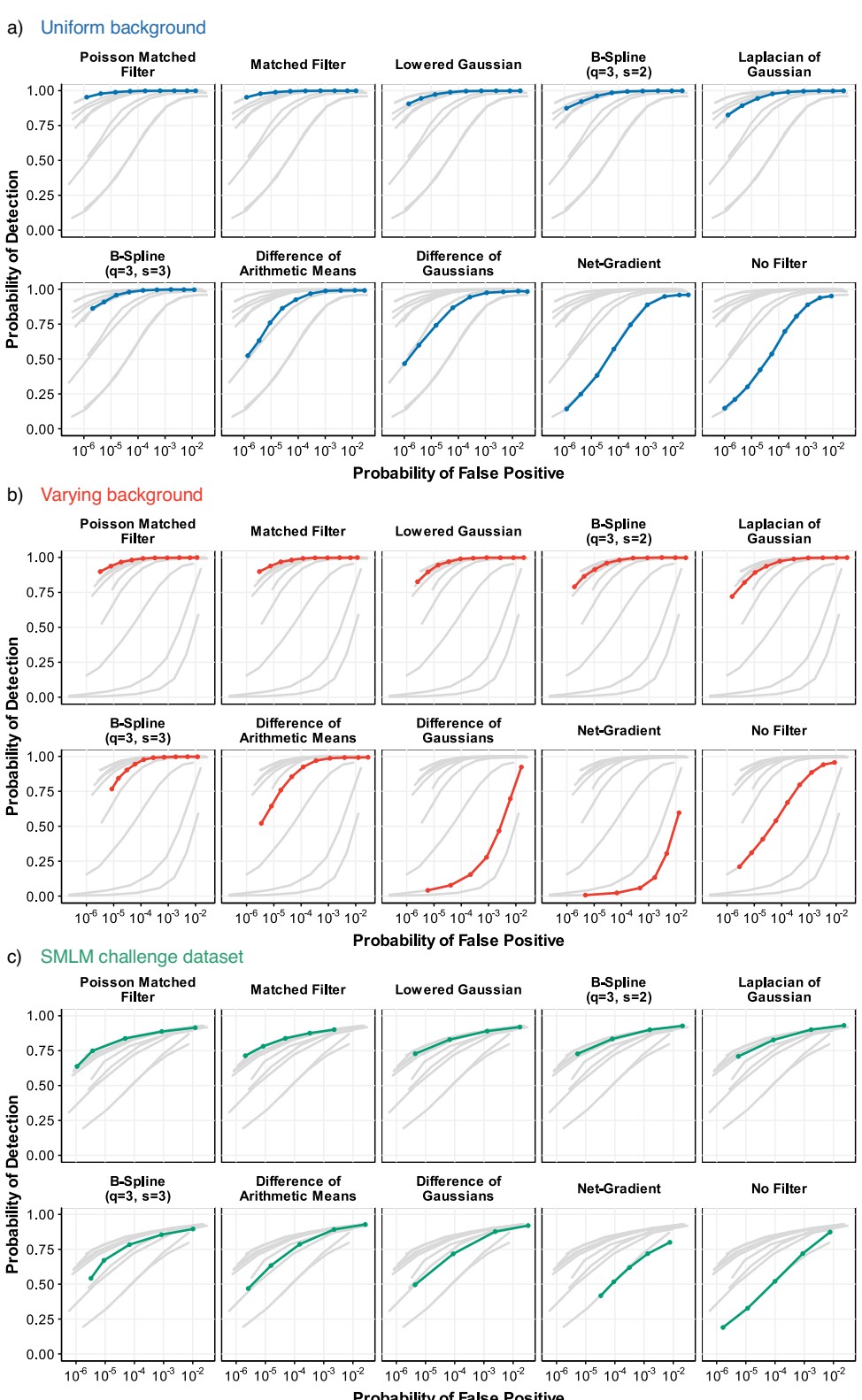

**Fig. 4 | Performance of multiple detection method on simulated data.** Performance evaluation by Receiver Operating Characteristic (ROC) curves showing the probability of detection $P_D$ as a function of false positive probability $P_{FP}$ for various methods across three different scenarios **a** Uniform background (blue): The Poisson Matched Filter (PMF) and Matched Filter (MF) consistently achieve the highest $P_D$ for any given $P_{FP}$, outperforming other methods. **b** Varying background (red): The performance difference becomes more pronounced under varying background conditions. The PMF maintains its superior detection capability. **c** SMLM challenge dataset[5] (green). The SMLM challenge dataset provides a realistic test scenario, where the PMF and MF continue to outperform other methods.

**Table 1 | List of detection filters and availability in SMLM software**

| Filters | Acronyms | References | Software |
|---|---|---|---|
| A trous B-spline wavelet | BS | 9,52 | ThunderStorm[61], SMAP[55], |
| Difference of Arithmetic means | DoA | 53,54 | ThunderStorm[61] |
| Difference of Gaussians | DoG | 55–57 | ThunderStorm[61], SMAP[55], QuickPALM[56], TrackMate[58] |
| Laplacian of Gaussian | LoG | 8,58 | TrackMate[58] |
| Lowered Gaussian | LG | 59 | DAOSTORM[59] |
| Matched Filter | MF | 22,25,26 | SimpleSTORM[26] |
| Net-gradient search | Net-grad | 60 | Picasso[60] |

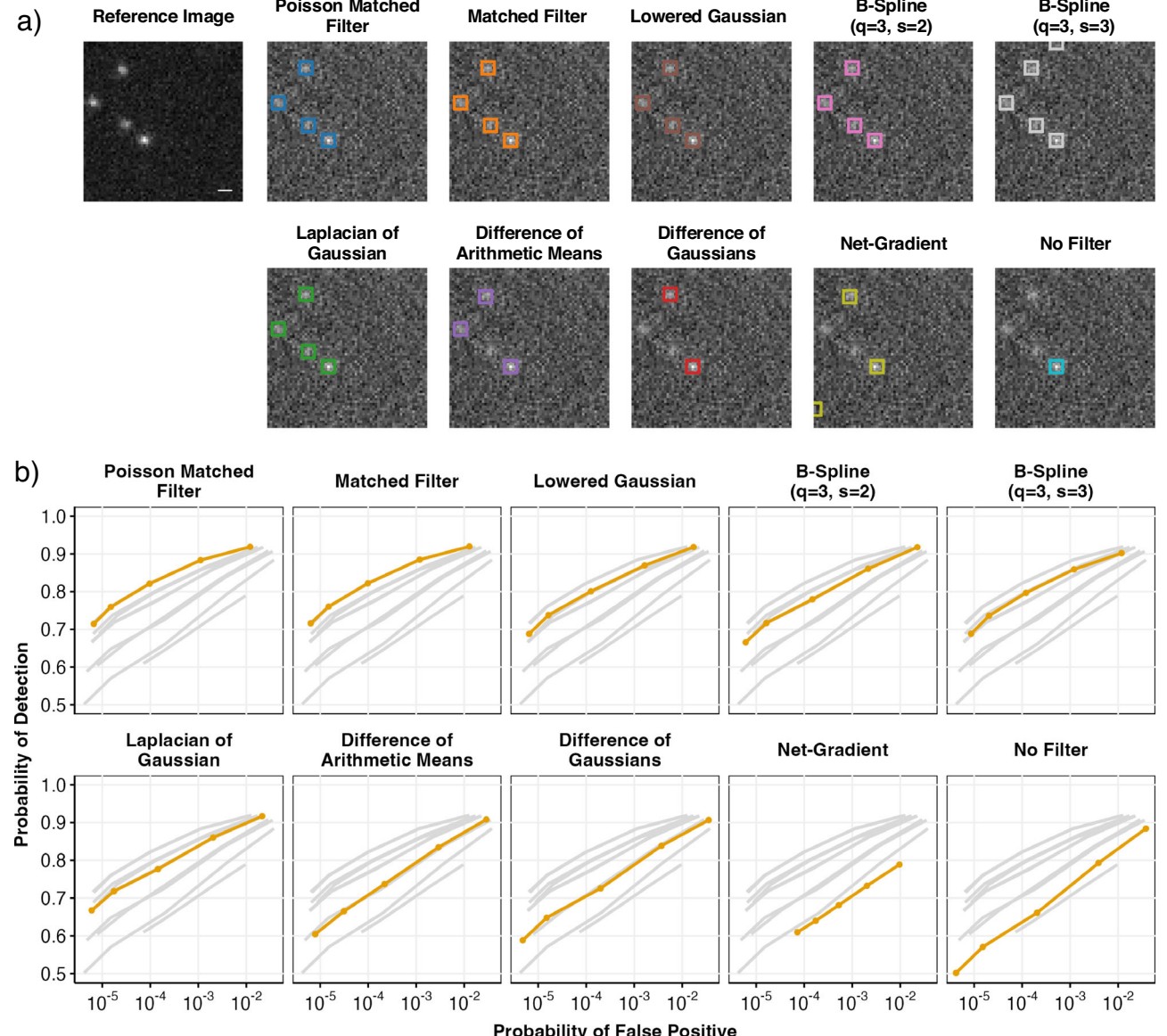

**Fig. 5 | Performance evaluation of multiple detection filtering methods over experimental dataset. a** Examples of detected gold nanoparticles in images with extra added light source compared to the reference image without interference. Scale bar is 500 nm. **b** Receiver operating characteristics showing the probability of detections $P_D$ of selected methods as a function of false positive probability $P_{FP}$ on the experimental dataset. Results show the top performance of the PMF and MF filters.

## Computational efficiency

We have designed a computationally efficient implementation of the PMF and probabilistic thresholding. The main demand for computational resources arises from the estimation of unknown parameters in GLRT. We present a series of simplifications and approximations that lead to a computationally efficient implementation in the form of linear filtering. The primary simplification lies in estimating the molecule's unknown position ($m_0$, $n_0$), where we restrict the potential values to integers, resulting in a negligible performance degradation (see Supplementary Fig. 15), but allowing much faster calculations.

Additionally, we simplify the estimation of the filtering parameters ($a$, $b$, and $\sigma$) for the PMF. Fixed values for parameters $a$, $b$ can be pre-determined through molecule localization over a small set of frames within the SMLM software, without requiring user interaction. This step becomes unnecessary when employing matched filter, as it is independent of $a$ and $b$. We assume that the last parameter, $\sigma$, is known based on the measurement setup.

In terms of computational complexity, our analysis (Supplementary Note 10) shows that the PMF operates as a linear filter, with an asymptotic complexity no worse than that of other common filtering methods. The adaptive background estimation for probabilistic thresholding has similar or not worse complexity as the linear filtering methods, making it computationally efficient for real-time applications. The computational complexity analyses are complemented by examples of specific run times on our regular desktop PC. We present a straightforward implementation of proposed methods using standard Python libraries (details in Supplementary Note 2).

When comparing the numerical complexity in the detection part of the SMLM algorithm to the Deep Learning (DL) methods, the DL segmentation is more numerically challenging than the linear filtering and thresholding presented in our method. Typical U-net-based[38] DL segmentation algorithms contain an image encoder that consists of a parallel bank of convolutional filters, i.e., consisting of many linear filtering operations. The computational complexity of current-state-of-the-art DL segmentation models is far exceeding linear filtering. For example, the number of parameters in the segment anything model[39] alone is over 600 million. DL models are typically less demanding and faster than the analytical solution if it is a full model that represents the entire chain of SMLM processing operations including the fitting of sub-pixel positions, the most computationally demanding part of the algorithm[40,41]. See Supplementary Note 11 for further comparison of our method and deep learning.

In summary, we show that matched filter paired with background-adaptive probabilistic thresholding is the optimal choice for analyzing SMLM data. This resolves issues related to parameter configurations by users and reproducibility of results while allowing to address the impact of false positive detections on subsequent qualitative localization analysis, for example clustering analysis, drift correction, localization dynamics and others. We advocate for adopting matched filtering and adaptive probabilistic thresholding as standard features in SMLM software. Beyond SMLM, our methodology holds potential benefits for other imaging techniques suffering from low SNR.

## Methods

### Neyman–Pearson's Likelihood Ratio Test (LRT)
The Neyman-Pearson theorem states that the likelihood ratio test

$$\Lambda(\mathbf{R}) = \frac{p(\mathbf{R}|H_1)}{p(\mathbf{R}|H_0)} \overset{H_1}{\underset{}{>}} \tau \tag{6}$$

offers a theoretically optimal molecule detection performance. The word optimal in this context means achieving the maximum probability of detection $P_D$ while keeping the probability of false positive $P_{FP}$ equal to a selected level[16]. For a list of synonymous terminology, we refer to Supplementary Table 5. The property of maintaining a constant probability of false positive $P_{FP}$ is often referred to as constant false alarm rate detection, especially in the context of radar signal processing[18].

The symbol of inequality in test (6) means that we reject the null hypothesis $H_0$ (signal absent) and accept the alternative hypothesis $H_1$ (signal present) if the inequality is fulfilled. For the opposite inequality, we do not reject the signal-not-present hypothesis $H_0$. The symbol $\tau$ denotes the detection threshold, which is determined by the chosen false positive detection probability $P_{FP}$, as presented in "Probabilistic Thresholding and Probability of Detection".

The derivation of a PMF filter can be shown in an arbitrary number of dimensions, but for simplicity we consider a 2D data space where symbol $\mathbf{R}$ is a raw image matrix $\mathbf{R} \in \mathbb{N}^{M \times N}$ of the size $M \times N$. Symbol $p(\mathbf{R}|H_1)$ denotes the conditional probability density function (denoted as likelihood function) of obtaining $\mathbf{R}$ given the hypothesis signal present. Similarly, likelihood $p(\mathbf{R}|H_0)$ is conditioned by the hypothesis signal absent.

### Statistical hypotheses of image formation model
Suppose that we have a PSF signal model in Poisson noise-limited conditions. This model corresponds to the scenario where all other types of noise are negligible, and what remains is the irremovable photon noise arising from the particle nature of the light[42]. At the same time, we assume precise conversion from analog-digital units to photon counts, using a well-calibrated camera gain and baseline offset[26,43]. Then, the hypothesis of signal absent $H_0$ and present $H_1$ has the following statistical description

$$\begin{aligned} H_0 &: \mathbf{R} \in \mathbb{N}^{M \times N}, [\mathbf{R}]_{mn} = r_{mn} \sim \mathscr{P}(b), \\ H_1 &: \mathbf{R}, [\mathbf{R}]_{mn} = r_{mn} \sim \mathscr{P}(as_{m-m_0, n-n_0}(\sigma) + b), \end{aligned} \tag{7}$$

where letter $\mathscr{P}(b)$ denotes the Poisson distribution with the mean $b$. Symbol $\sim$ means "distributed as". The $mn$th pixel of raw image $[\mathbf{R}]_{mn}$ is denoted by the lowercase symbol $r_{mn}$. Other symbols indicate the number of emitted photons $a$, the mean background level $b$, and the sampled PSF $s_{mn}(\sigma)$. Any PSF model is applicable here, such as the 2D Gaussian model

$$s_{mn}(\sigma) = \frac{1}{2\pi\sigma^2} e^{-\frac{1}{2\sigma^2}(m^2 + n^2)}, m \in \mathbb{Z}, n \in \mathbb{Z}, \tag{8}$$

the integrated Gaussian model or others including the measured one. Symbol $\sigma$ denotes the standard deviation of PSF. Unknown coordinates of the molecule $m_0, n_0$ are assumed to fit the size of the raw image with a tolerance of $K$ pixels given by the expected size of the PSF.

### Parametric likelihood functions
The likelihood functions for both hypotheses (7) are obtained as the joint distribution of independent identically distributed Poisson random variables as

$$p(\mathbf{R}|H_0, \boldsymbol{\theta}_0) = \prod_{mn} \frac{b^{r_{mn}}}{r_{mn}!} e^{-b}, \tag{9}$$

$$p(\mathbf{R}|H_1, \boldsymbol{\theta}_1) = \prod_{mn} \frac{\left(as_{m-m_0, n-n_0}(\sigma) + b\right)^{r_{mn}}}{r_{mn}!} e^{-(as_{m-m_0, n-n_0}(\sigma) + b)}, \tag{10}$$

where we have shortened the notation by symbol $\prod_{mn} \triangleq \prod_{m=0}^{M-1} \prod_{n=0}^{N-1}$. Symbol $\triangleq$ means the equality from the definition. We see that the likelihood functions (9), (10) cannot be evaluated because they depend on the unknown parameters denoted by vectors $\boldsymbol{\theta}_0 = b$, $\boldsymbol{\theta}_1 = [a, b, \sigma, m_0, n_0]^T$, where $(\star)^T$ is the matrix transposition. Therefore, the likelihoods cannot be directly used in the LRT (6).

### Generalized Likelihood Ratio Test (GLRT)
GLRT accepts hypothesis $H_1$, if the likelihood ratio is greater than the threshold $\tau$

$$\Lambda_G(\mathbf{R}) = \frac{p(\mathbf{R}|H_1, \hat{\boldsymbol{\theta}}_1)}{p(\mathbf{R}|H_0, \hat{\boldsymbol{\theta}}_0)} \overset{H_1}{\underset{}{>}} \tau \tag{11}$$

where the unknown parameters in $\boldsymbol{\theta}_0$ and $\boldsymbol{\theta}_1$ have been estimated by the maximum likelihood estimation

$$\hat{\boldsymbol{\theta}}_i = \underset{\boldsymbol{\theta}_i}{\operatorname{argmax}}\, p(\mathbf{R}|H_1, \boldsymbol{\theta}_i), \quad i \in \{1, 2\}, \tag{12}$$

where the estimated values are labeled by a hat symbol $\hat{\star}$. The GLRT is adjusted for the likelihoods (9), (10) to the inequality

$$\Lambda_G(\mathbf{R}) = \prod_{mn} \left(1 + \frac{\hat{a}}{\hat{b}} s_{m-\hat{m}_0, n-\hat{n}_0}(\hat{\sigma})\right)^{r_{mn}} e^{-\hat{a} s_{m-\hat{m}_0, n-\hat{n}_0}(\hat{\sigma})} \overset{H_1}{\underset{}{>}} \tau. \tag{13}$$

The inequality in (13) does not change if we take the logarithm of both sides, leading to

$$\ln \Lambda_G(\mathbf{R}) = \sum_{mn} r_{mn} \ln\left(1 + \frac{\hat{a}}{\hat{b}} s_{m-\hat{m}_0, n-\hat{n}_0}(\hat{\sigma})\right) - \hat{a} \overset{H_1}{\underset{}{>}} \ln \tau, \tag{14}$$

where we assume that PSF is normalized such that $\sum_{m,n} s_{m-\hat{m}_0, n-\hat{n}_0}(\hat{\sigma}) = 1$. Here, symbol $\sum_{mn}$ means $\sum_{mn} \triangleq \sum_{m=0}^{M-1} \sum_{n=0}^{N-1}$.

### Poisson Matched Filter (PMF)

Once the parameters in (14) are estimated, we can further adjust the detection inequality to the form

$$T(\mathbf{R}) = \sum_{mn} r_{mn} w_{m-\hat{m}_0, n-\hat{n}_0}(\hat{a}, \hat{b}, \hat{\sigma}) \overset{H_1}{\underset{}{>}} \tau', \tag{15}$$

where $T(\mathbf{R})$ is called a test statistic. Symbol $\tau' = \ln \tau + \hat{a}$ is a new decision threshold incorporating estimate $\hat{a}$, the value of which is determined by the desired probability of false positive $P_{FP}$. Left-hand side of the inequality (15) is a weighted arithmetic mean with the weights given by

$$w_{mn}(a, b, \sigma) = \ln\left(1 + \frac{a}{b} s_{mn}(\sigma)\right). \tag{16}$$

We call the weights in (16) the filter kernel of the PMF. To simplify the notation, we omit the functional dependency on the parameters and simply write $w_{mn}$. We assume that the PMF kernel takes significant non-zero values in the interval $[-L, L] \times [-L, L]$ and describe the PMF kernel by matrix $\mathbf{W} \in \mathbb{R}^{K \times K}$, $K = 2L + 1$, where $[\mathbf{W}]_{mn} = w_{mn}$. We see that the PMF shape is non-linearly dependent on PSF width $\sigma$ and on the signal-to-background ratio $a/b$.

### Estimation of unknown parameters in GLRT: implementation as a linear filter

To efficiently implement PMF as a linear filter, we present a series of simplifications and approximations.

**Estimation of unknown position.** The first and fundamental simplification is obtained when we approximate the unknown position of the molecule $m_0, n_0 \in \mathbb{R}$ to acquire only integer values $m_0, n_0 \in \mathbb{Z}$, where symbols $\mathbb{R}$ and $\mathbb{Z}$ refer to the real and integer numbers. Let us consider for now GLRT (11), where all unknown parameters except of the molecule position $m_0, n_0$ are known

$$\Lambda_G(\mathbf{R}) = \max_{m_0, n_0} \frac{p(\mathbf{R}|H_1, \hat{a}, \hat{b}, \hat{\sigma}, m_0, n_0)}{p(\mathbf{R}|H_0, \hat{b})} \tag{17}$$

which can be adjusted (using the even symmetry of $w_{mn}$) to the form

$$T(\mathbf{R}) = \max_{m_0, n_0} \sum_{mn} r_{mn} w_{m_0-m, n_0-n} = \sum_{mn} r_{mn} w_{\hat{m}_0-m, \hat{n}_0-n}, \tag{18}$$

where

$$\hat{m}_0, \hat{n}_0 = \underset{m_0, n_0}{\operatorname{argmax}} \sum_{mn} r_{mn} w_{m_0-m, n_0-n}. \tag{19}$$

Now, if $m_0, n_0 \in \mathbb{Z}$, we recognize in (18) a discrete 2D convolution of raw image $r_{mn}$ and PMF filter $w_{mn}$ (16). The GLRT then takes the form

$$T(\mathbf{R}) = \max_{m_0, n_0} [r * * w]_{m_0 n_0} = [r * * w]_{\hat{m}_0 \hat{n}_0} \overset{H_1}{\underset{}{>}} \tau', \tag{20}$$

where symbol ** denotes a discrete 2D convolution. We denote the filtered image by matrix $\mathbf{T}$, where $[\mathbf{T}]_{mn} = (r^{**}w)_{mn}$. The maximum of filtered image $[\mathbf{T}]_{\hat{m}_0 \hat{n}_0}$ is compared to the threshold to decide on the presence of a molecule. Note that the derivation naturally assumes a single-molecule detection problem. For multiple non-overlapping molecule detections, the maximum in (20) needs to be replaced by the local maximum in the neighborhood of the PSF size. Reference[44] summarizes various efficient ways to implement local maximum search. Note that the non-maximum suppression does not need to be evaluated for all pixels of the image, but only for those that are larger than the detection threshold $\tau'$. The generalization of the detection problem of multiple overlapping molecules (denoted as crowded field problem) is given in Supplementary Note 12.

**Estimation of parameters $a$, $b$, and $\sigma$.** The PMF filter kernel $w_{mn}$ (16) then depends on the unknown molecule parameters $a$, $b$, and PSF parameter $\sigma$. Some references[19,25] assume knowledge of the PSF $\sigma$ parameter or at least its roughly estimated value[45] given e.g., by the expression $\hat{\sigma} = 0.21 \lambda_{em}/NA$, where $\lambda_{em}$ is the emission wavelength and NA is the numerical aperture. The maximum likelihood estimate of $a$ and $b$ with fixed $\hat{\sigma}$ is a convex optimization problem with effective algorithms available[46]. However, the estimation needs to be evaluated for each pixel in the raw data which is still numerically too intensive.

To implement PMF as a linear filter, we need to fix the values of the unknown parameters $a$, $b$. A possible approach takes advantage of the fact that the detection algorithm is part of SMLM software. We use the 0.25 percentile of the true parameter values obtained from the SMLM localization algorithm applied on a randomly selected set of a few frames of the raw data. This approach conveniently does not require user interaction.

### Probabilistic thresholding and probability of detection

We take advantage of the knowledge of the analytical closed-form formula for the probability of false positive $P_{FP}$. We evaluate two performance metrics (probability of false positive $P_{FP}$ and probability of detection $P_D$) characterizing two fundamental error types of False Positive (FP) and False Negative (FN) detection. The probability threshold is then obtained by inverting the analytical expression for $P_{FP}$.

**Probability of false positive detection.** Probability $P_{FP}$ is the probability that the GLRT (15) decides in favor of $H_1$ (test statistic $T(\mathbf{R})$ is higher than threshold $\tau'$) but $H_0$ is true

$$P_{FP} \triangleq \Pr\left[\hat{H} = H_1 | H_0\right] = \Pr\left[T(\mathbf{R}) > \tau' | H_0\right], \tag{21}$$

where the symbol $\hat{H}$ denotes the chosen hypothesis. We determine the performance by analyzing the distribution of the test statistics $T(\mathbf{R})$ on the condition that the molecule position coordinates are integer numbers $m_0, n_0 \in \mathbb{Z}$. Let us re-write the weighted sum of random variables (15) using the fact that PMF is significantly non-zero for

$m, n \in [-L, L]$ to get

$$T(\mathbf{R}) = \sum_{m,n=-L}^{L} r_{m+\hat{m}_0, n+\hat{n}_0} w_{m,n} \tag{22}$$

where we have shortened the notation by $\sum_{m,n=-L}^{L} \triangleq \sum_{m=-L}^{L} \sum_{n=-L}^{L}$. The test statistic (22) is a random variable with a distribution well approximated by a Gaussian distribution due to the central limit theorem. The full statistical description is obtained by expressing the first two moments for the null hypothesis $H_0$ where $r_{m+\hat{m}_0, n+\hat{n}_0} \sim \mathscr{P}(b)$ and so

$$E[T(\mathbf{R})|H_0] = E\left[\sum_{m,n=-L}^{L} r_{m+\hat{m}_0, n+\hat{n}_0} w_{mn}|H_0\right] = b \sum_{m,n=-L}^{L} w_{mn}, \tag{23}$$

$$\mathrm{var}[T(\mathbf{R})|H_0] = \mathrm{var}\left[\sum_{m,n=-L}^{L} r_{m+\hat{m}_0, n+\hat{n}_0} w_{mn}|H_0\right] = b \sum_{m,n=-L}^{L} w_{mn}^2, \tag{24}$$

using the linearity of the mean operator $E[\star]$ and the rule for the variance of the sum of independent random variables. The false positive probability $P_{FP}$ can be expressed in a closed form using the complementary cumulative distribution function (also denoted as the survival function) of standard normal distribution

$$Q(x) \triangleq \frac{1}{\sqrt{2\pi}} \int_x^\infty \exp\left(-\frac{t^2}{2}\right) dt \tag{25}$$

as

$$P_{FP} = Q\left(\frac{\tau' - E[T(\mathbf{R})|H_0]}{\sqrt{\mathrm{var}[T(\mathbf{R})|H_0]}}\right) = Q\left(\frac{\tau' - b\sum_{m,n=-L}^{L} w_{mn}}{\sqrt{b\sum_{m,n=-L}^{L} w_{mn}^2}}\right). \tag{26}$$

**Probability of detection.** Similarly as for $P_{FP}$, the detection probability is given by

$$P_D \triangleq \Pr\left[\hat{H} = H_1|H_1\right] = \Pr\left[T(\mathbf{R}) > \tau'|H_1\right]. \tag{27}$$

We can express $P_D$ in a close form when the integer molecule position is estimated correctly ($\hat{m}_0 = m_0, \hat{n}_0 = n_0$) where the received signal $r_{m+\hat{m}_0, n+\hat{n}_0} \sim \mathscr{P}(a s_{mn} + b)|H_1$ and so the test statistics are

$$E[T(\mathbf{R})|H_1] = E\left[\sum_{m,n=-L}^{L} r_{m+\hat{m}_0, n+\hat{n}_0} w_{mn}|H_0\right]$$
$$= a \sum_{m,n=-L}^{L} s_{mn} w_{mn} + b \sum_{m,n=-L}^{L} w_{mn}, \tag{28}$$

$$\mathrm{var}[T(\mathbf{R})|H_1] = \mathrm{var}\left[\sum_{m,n=-L}^{L} r_{m+\hat{m}_0, n+\hat{n}_0} w_{mn}|H_0\right] = a \sum_{m,n=-L}^{L} s_{mn} w_{mn}^2 + b \sum_{m,n=-L}^{L} w_{mn}^2. \tag{29}$$

The detection probability is then

$$P_D = Q\left(\frac{\tau' - E[T(\mathbf{R})|H_1]}{\sqrt{\mathrm{var}[T(\mathbf{R})|H_1]}}\right) = Q\left(\frac{\tau' - a\sum_{m,n=-L}^{L} s_{mn} w_{mn} - b\sum_{m,n=-L}^{L} w_{mn}}{\sqrt{a\sum_{m,n=-L}^{L} s_{mn} w_{mn}^2 + b\sum_{m,n=-L}^{L} w_{mn}^2}}\right). \tag{30}$$

**Probabilistic threshold controlling the false positive probability.** Probability $P_{FP}$(26) depends on the mean background level $b$ and the threshold $\tau'$. After capturing the image, the mean background parameter $b$ can be estimated considering that the quantity of background

pixels is usually significantly higher than that of signal pixels (more discussed in "Background Estimation for Adaptive Thresholding"). Then, the only parameter controlling the probability $P_{FP}$ is the decision threshold $\tau'$. The $Q$ function (25) is monotonic, so it has an inversion (sometimes called inverse survival function) leading to the expression of the decision threshold as

$$\tau' = \sqrt{\hat{b} \sum_{m,n=-L}^{L} w_{mn}^2 Q^{-1}(P_{FP}) + \hat{b} \sum_{m,n=-L}^{L} w_{mn}}. \tag{31}$$

We refer to the threshold (31) as the probabilistic threshold to emphasis that the threshold is determined by the selected probability of false positive $P_{FP}$. Note that the probabilistic threshold (31) is not conditioned by the PMF filter and can be used with any other linear filter common in SMLM.

## Background estimation for adaptive thresholding

We assume a sufficiently sparse fluorescence signal with most of the pixels being the background that allows a maximum likelihood estimation[47] of the parameters of the Poisson and normal distribution necessary to calculate the probabilistic thresholds (31). Estimating the local background level around the pixel under evaluation by the detection algorithm allows to adapt the threshold to slow-varying background.

**Mean estimate.** The optimal maximum likelihood estimation of the central moment for both Poisson and Gaussian distributions is given by the arithmetic mean[21,47]. Let us denote the local arithmetic mean background estimate as

$$\hat{b}_{ij} = \frac{1}{\sum_{mn} w_{mn}} \operatorname*{mean}_{(m,n) \in \mathscr{K}_{ij}} [t_{mn}], \tag{32}$$

where $t_{mn}$ denotes the $mn$th pixel of the filtered image $t_{mn} = [\mathbf{T}]_{mn} = (r^{**}w)_{mn}$. The normalization by the sum of weights $\sum_{mn} w_{mn}$ corresponds to (23). The symbol $\mathscr{K}_{ij}$ describes the set of indexes centered at position $(i, j)$ corresponding to the reference pixels used in the local arithmetic mean as shown in Fig. 3d). The local mean (32) can be efficiently implemented as a linear filter as

$$\hat{b}_{ij} = \frac{1}{\sum_{mn} w_{mn}} (t * k)_{ij} = \frac{1}{\sum_{mn} w_{mn}} \sum_{m,n=-L}^{L} k_{mn} \cdot t_{i-m, j-n}, \tag{33}$$

where $\mathbf{K} \in \mathbb{R}^{K \times K}$ denotes the kernel matrix with $k_{mn}$ being the $mn$th value $[\mathbf{K}]_{mn} = k_{mn}$ and $K$ is the kernel size assumed to be an odd number with $K = 2L + 1$. The kernel is normalized to perform the arithmetic mean so $\sum_{mn} k_{mn} = 1$. For example,

$$\mathbf{K} = \frac{1}{40} \begin{bmatrix} 1 & 1 & 1 & 1 & 1 & 1 & 1 \\ 1 & 1 & 1 & 1 & 1 & 1 & 1 \\ 1 & 1 & 0 & 0 & 0 & 1 & 1 \\ 1 & 1 & 0 & 0 & 0 & 1 & 1 \\ 1 & 1 & 0 & 0 & 0 & 1 & 1 \\ 1 & 1 & 1 & 1 & 1 & 1 & 1 \\ 1 & 1 & 1 & 1 & 1 & 1 & 1 \end{bmatrix} \tag{34}$$

is the square-shaped kernel of size 7 with a guard interval of 3 pixels. In the case where nearby fluorescence frequently occurs so that the signal is often present in the reference pixels, the arithmetic mean (32) is biased and better results can be obtained by local median filtering

$$\hat{b}_{ij} = \frac{1}{\sum_{mn} w_{mn}} \operatorname*{median}_{(m,n) \in \mathscr{K}_{ij}} [t_{mn}]. \tag{35}$$

There are known efficient algorithms for median filtering with linear[48], and even constant[49] computational complexity.

**Variance estimate.** Probabilistic thresholding for Poisson-Gaussian noise (described in Supplementary Note 4) also requires estimation of the second moment, which can be done as local filtering. The variance estimation of the normal distribution can be implemented by linear mean filter as

$$\hat{\sigma}_{b,ij}^2 = \frac{1}{\sum_{mn} w_{mn}^2} \operatorname*{mean}_{(m,n)\in\mathcal{K}_{ij}} \left[ \left( t_{mn} - \hat{b}_{mn} \sum_{mn} w_{mn} \right)^2 \right]. \qquad (36)$$

Robust estimation can be performed e.g., by the Inter-Quantile Range estimate as

$$\hat{\sigma}_{b,ij}^2 = \frac{1}{\sum_{mn} w_{mn}^2} 0.7413 \left( \operatorname*{iqr}_{(m,n)\in\mathcal{K}_{ij}} \left[ t_{mn} \right] \right), \qquad (37)$$

or by median absolute deviation and other alternatives[50,51].

### Reference methods
We consider the following filters as reference methods (in alphabetical order): B-Spline (BS) wavelet filter[9,52], DoA means[53,54], DoG[55–57], LoG[8,58], LG[59], MF[25,26], and the net-gradient search[60]. All methods except the net-grad search are based on linear filtering and for their description it is sufficient to define just the filter kernel. The entire detection procedure for the general method using a linear filter is identical to that of PMF: the test statistic is calculated using Eqs. (15) and (20), the probabilistic decision threshold using (2) in the manuscript, and (14) in Supplementary Note 4, and the background estimation (Background estimation for adaptive thresholding). Selected filters are visualized in Supplementary Note 13.

### À trous B-Spline (BS) wavelet filter.
Continuous B-Spline basis is given by the recursion

$$B_1(x) = \begin{cases} 1 & 0 \le x < 1, \\ 0 & \text{else,} \end{cases} \qquad (38)$$

$$B_q(x) = \frac{x}{q-1} B_{q-1}(x) + \frac{q-x}{q-1} B_{q-1}(x-1), \qquad (39)$$

where $q \in \mathbb{N}$ is the BS order. Scaled, normalized, and continuous BS basis with the scale $s \in \mathbb{N}$ is

$$BS(x,q,s) = \frac{1}{s} B_q\left(\frac{x}{s} + \frac{q}{2}\right). \qquad (40)$$

The sampled BS wavelet for the number of samples $K = 2\lceil qs/2 \rceil - 1$ is usually called $\mathbf{k}_1 \in \mathbb{R}^L$ so

$$\mathbf{k}_1(q,s) = BS(i,q,s), \; i \in \{-L, -L+1, \dots, L\}, \qquad (41)$$

where $K = 2L+1$. For example $\mathbf{k}_1(3,2) = (1/16, 1/4, 3/8, 1/4, 1/16)$. The à trous vector $\mathbf{k}_2 \in \mathbb{R}^{2K-1}$ denotes the vector $\mathbf{k}_1$ with inserted in between zeros, such that $\mathbf{k}_2(3,2) = (1/16, 0, 1/4, 0, 3/8, 0, 1/4, 0, 1/16)$. We construct the filter kernel $[\mathbf{W}]_{mn} = w_{mn}$ of à trous BS wavelet as

$$\mathbf{W} = \mathbf{M}_1 ** (\Delta - \mathbf{M}_2), \qquad (42)$$

where $\mathbf{M}_1 = \mathbf{k}_1 \mathbf{k}_1^T \in \mathbb{R}^{K \times K}$ and $\mathbf{M}_2 = \mathbf{k}_2 \mathbf{k}_2^T \in \mathbb{R}^{(2K-1)\times(2K-1)}$ are matrices of outer products and $\Delta \in \mathbb{R}^{(2K-1)\times(2K-1)}$ is the zero matrix with a single 1 in the center $[\Delta]_{mn} = 1$, if $m = n = K$.

**Difference of Arithmetic means (DoA) filter.** The DoA filter is a two-parameter filter

$$w(x,y,a_1,a_2) = \frac{1}{a_1^2} \operatorname{rec}\left(\frac{x}{a_1}, \frac{y}{a_1}\right) - \frac{1}{a_2^2} \operatorname{rec}\left(\frac{x}{a_2}, \frac{y}{a_2}\right), \qquad (43)$$

where 2D rectangular function is defined as $\operatorname{rec}(x, y) = 1$ for $|x| \le \frac{1}{2}, |y| \le \frac{1}{2}$, and $\operatorname{rec}(x, y) = 0$ else. Reference[53] recommends parameters equal to $a_1 = 2\sigma + 1, a_2 = 2(2\sigma + 1)$ which we assume here. Symbol $\sigma$ is standard deviation of PSF.

**Difference of Gaussians (DoG) filter.** The DoG filter is a two-parameter filter

$$w(x,y,\sigma_1,\sigma_2) = \frac{1}{2\pi\sigma_1^2} e^{-\frac{1}{2\sigma_1^2}(x^2+y^2)} - \frac{1}{2\pi\sigma_2^2} e^{-\frac{1}{2\sigma_2^2}(x^2+y^2)}, \qquad (44)$$

which can approximate the LoG filter[12] very-well for $\sigma_1 = 0.805\sigma_{LoG}, \sigma_2 = 1.288\sigma_{LoG}$. SMLM literature[56] recommends values $\sigma_1 = 0.5\sigma, \sigma_2 = 2 \text{ FWHM} = 4.710\sigma$ which we assume here.

**Laplacian of Gaussian (LoG) filter.** The LoG filter also known as Mexican Hat, 2nd Derivative of Gaussian, or Ricker Wavelet, is given by

$$w(x,y,\sigma_{LoG}) = \left( \frac{2\sigma_{LoG}^2 - x^2 - y^2}{2\pi\sigma_{LoG}^6} \right) e^{-\frac{1}{2\sigma_{LoG}^2}(x^2+y^2)}, \qquad (45)$$

where LoG parameter $\sigma_{LoG}$ is assumed to closely approximate Gaussian function which leads to $\sigma_{LoG} = \sqrt{2}\sigma$ with $\sigma$ being standard deviation of PSF.

**Lowered Gaussian (LG) filter.** The LG filter has a Gaussian kernel shape (8) from which the total value is subtracted such that the filter has the sum of values equal to zero $\sum_{m,n} w_{m,n} = 0$. The zero-sum property (also found in LoG, DoG, DoA, and BS filters) is advantageous because it implies that a slowly changing background is filtered out.

**Matched Filter (MF).** MF has the continuous form of filter kernel equal to $s(\sigma)$ PSF model, which e.g., for Gaussian PSF model equals to

$$w(x,y,\sigma) = \frac{1}{2\pi\sigma^2} e^{-\frac{1}{2\sigma^2}(x^2+y^2)} \qquad (46)$$

with $\sigma$ being s.t.d. of PSF function.

**Net-gradient search.** The net-gradient search is a non-linear filtering method that sums raw image gradient vector magnitudes in a rectangle box. We implement the image gradient by Sobel filters

$$\mathbf{G} = \begin{bmatrix} 1 & 0 & -1 \\ 2 & 0 & -2 \\ 1 & 0 & -1 \end{bmatrix}, \quad \mathbf{H} = \begin{bmatrix} 1 & 2 & 1 \\ 0 & 0 & 0 \\ -1 & -2 & -1 \end{bmatrix} \qquad (47)$$

The squared outputs of these filters are summed and then the square root is taken to compute the magnitude of the gradient at each pixel. These gradient magnitudes are then averaged using a $5 \times 5$ box filter to smooth the final result.

**Poisson Matched Filter (PMF).** The PMF filter derived in "Poisson Matched Filter (PMF)" has the continuous form of filter kernel equal to

$$w(x,y,a,b,\sigma) = \ln\left(1 + \frac{a}{b} s(\sigma)\right), \qquad (48)$$

where the procedure for selection of parameters $a$, $b$, $\sigma$ is discussed in "Estimation of Unknown Parameters in GLRT: Implementation as a Linear Filter" and $s(\sigma)$ is the sampled PSF function.

**No filter.** As a reference method, we also consider the no-filter case, which can be described as a linear filter with discrete weights $w_{mn} = 1$ for $m = n$, and $w_{mn} = 0$ else.

### Performance evaluation

**Simulations with uniform background and Poisson noise.** First, the detection methods are tested in a constant background environment where a single randomly placed spot in Poisson noise is detected as illustrated in Supplementary Fig. 10.

In every Monte Carlo run, the spot position is a different real random vector. True Positive (TP) is counted when the detected position is within the tolerated distance of $\tau_{TP} = \sqrt{2}$ px from the ground true value. True Negative (TN) is counted as a correctly detected background pixel. False positive is a falsely marked detection which is farther than $\tau_{TP}$. False Negative (FN) is missed detection. By repeating the experiment many times (we ran $M_{MC} = 5000$ Monte Carlo trials), we can express $\hat{P}_D = \text{mean}\left[\text{TP}/\text{TP}+\text{FN}\right]$ and $\hat{P}_{FP} = \text{mean}\left[\text{FP}/\text{FP}+\text{TN}\right]$.. We select values of low signal-to-background ratio $a/b = 0.75$ (particularly with $a = 750$ photons and $b = 1000$ photons) where the differences between the methods are more pronounced. In cases with medium to high $a/b$ ratios, the differences between the methods are smaller. The size of the raw image is $128 \times 128$ pixels, s.t.d. of PSF is $\sigma = 1$ px and the non-maximum suppression box is a $5 \times 5$ box. All the considered methods use probabilistic thresholding (introduced in "Probabilistic Thresholding and Probability of Detection") to guarantee selected $P_{FP}$, but we rather use the estimated $\hat{P}_{FP}$ in the plots that are occasionally slightly different from the true values. This is the reason why the plot markers are not perfectly aligned. PMF used the precise values of the unknown parameters $a$, $b$, $\sigma$.

**Simulations with varying background and Poisson noise.** We assume a parabolic type of background to model a non-uniform illumination similar to the effect of vignetting[37] which is coarsely approximated by parabola ($\cos^4(\alpha) \simeq 1 - 2\alpha^2$). We use the parabola model $1/c^2(x^2 + y^2)$ with curvature value $c = 10$ pixels, which is then mean-shifted to have the same mean as in the case of uniform background (i.e., mean value $b = 1000$ ph). Supplementary Fig. 16 illustrates such an example signal.

PMF, MF, and no filter methods use local adaptive background estimation. The local adaptive background estimation used local mean implementation (32) in the floating square doughnut-shape kernel (illustrated in Fig. 3d) of the size $13 \times 13$ pixels with guard interval $5 \times 5$.

**Performance in bundled tubes long sequence (BT-LS) dataset.** We used the publicly available dataset provided by the 2013 SMLM software challenge[5]. This dataset has a very good signal-to-noise ratio therefore, the performance of the most tested algorithms is close to ideal. In order to better observe the differences between the methods in challenging conditions, we added a noisy background of the level of 4000 photons. Type of the noise in BT-LS data corresponds to the Poisson-Gaussian noise, so the probabilistic thresholding uses formula (14) in Supplementary Note 4 with background estimation (32) and (36).

**Performance in MT0_N2_LD dataset.** Microtubulin dataset MT0_N2_LD publicly provided by the 2016 SMLM software challenge[6] has a challenging signal-to-noise ratio and low density of emitters.

**Performance in experimental datasets.** To validate our method using experimental data, we created a reference sample with homogenously distributed single gold nanoparticles and visualized it in two modes. One under standard visualization conditions that result in bright diffraction-limited spots on a low background that are easy to detect and used this measurement as a reference. For the second visualization mode, we added an extra light source from the microscope lamp to increase the background and used it as testing data.

To reduce agglomeration of gold nanoparticles during deposition on a glass coverslip, we used gold nanoparticles that were attached to DNA origami (Supplementary Fig. 14) with biotinylated sites and deposited them on neutravidin-treated glass coverslips, previously passivated with BSA-biotin (Supplementary Fig. 13). A detailed description of the experimental dataset (used materials, nanoparticle functionalization, DNA origami synthesis, origami-particle hybridization, sample preparation, and measurement) is given in Supplementary Note 9.

We created two datasets. One with a medium level of interference light and the other with a strong interference light. Ground truth detections were obtained by evaluating the same ROIs but without interference light. In rare cases, some fluorophores were not present in the reference data, which slightly raised the false positive detection rate. More often, the weaker fluorescence emission in the reference data disappeared completely in the presence of interference or was below the background noise level, which then showed up as a missed detection in the evaluation.

The type of noise in the experimental data was close to Poisson-Gaussian noise, so the detection threshold was set as described in Supplementary Note 4 with background estimation (32) and (36).

## Data availability

All data are available at https://gitlab.com/VladkaPetrakova/petrakova_group/-/tree/main/mirek/publication/molecule_detection, data from the SMLM Challenges used for testing are available at https://bigwww.epfl.ch/smlm-datasets/challenge2D/datasets/Bundled_Tubes_Long_Sequence-GT/ and https://srm.epfl.ch/srm/dataset/challenge-3D-simulation/MT0.N2.LD/index.html.

## Code availability

The code is available at https://gitlab.com/VladkaPetrakova/petrakova_group/-/tree/main/mirek/publication/molecule_detection.

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

## Acknowledgements

This work was supported by Czech Science Foundation, grant number 21-17847M (M.H., N.H., S.F., V.P.) and Czech Academy of Sciences, Premie lumina quaeruntur LQ200402101 (K.U., D.R., V.P.). The authors acknowledge the assistance provided by the Research Infrastructure NanoEnviCz, supported by the Ministry of Education, Youth, and Sports of the Czech Republic under Project No. LM2023066. K.U. wants to acknowledge the support of CTU student grant SGS24/156/OHK4/3T/17. We thank Olga Pavlatová, Tomáš Chum, and Jakub Jungwirth for their valuable help.

## Author contributions

M.H. developed the theory, implementation, and simulations, D.R. and K.U. performed and set up measurements, N.H. and S.F. prepared samples and characterized them, V.P. acquired the funding and supervised the work. M.H. and V.P. wrote the manuscript with the contribution of all authors.

## Competing interests

The authors declare no competing interests.
