## [Transparent Peer Review file · Nature Communications]

Optimized Molecule Detection in Localization Microscopy with Selected False Positive Probability

Corresponding Author: Dr Vladimíra Petráková

Version 0:

Reviewer comments:

Reviewer #2

(Remarks to the Author)

This paper shows a new method for detection of single molecules in single molecule localization microscopy. The author's method (Poisson Matched Filter) incorporates a filter template with Poisson statistics, resulting in a better-performing method than previously available. The paper is well put together and is very thorough.

The authors should add the following reference, which shows a similar Monte-Carlo approach for performance evaluation of detection methods:

Křížek, et al, "Minimizing detection errors in single molecule localization microscopy," Opt. Express 19, 3226-3235 (2011).

Is it possible to evaluate more of the single molecule challenge data? Some of the data sets present in the challenge (2013 and 2016 data) have higher levels of noise, or higher densities of molecules.

In future work, it would be nice to see further imaging of DNA origami structures and also of structures within cells. This should show the authors method effectively reduces false molecule detection in real data.

To summarize I think this is a very good paper which should be published and it is a nice contribution to the field.

(Remarks on code availability)

Reviewer #3

(Remarks to the Author)

Concerns:

1. The method is well described mathematically, but its advantages over other methods are not sufficiently demonstrated. The graphics lack a comprehensive explanation, particularly regarding the quantification of comparative results in relation to other methods.
2. Introduction and abstract have to put the methodology into a broader context and emphasize on the importance of this result for the SMLM field. Additionally, it is important to highlight the advantages of the described method in comparison to new AI-based methods (e.g., BGnet, a deep neural network).
3. A computation speed comparison of the algorithms described is missing. Is this introduced algorithm suitable for real time applications which suffer from false signal detection?
4. As of Figure 4 the author mentioned that PMF & MF are the best filters for the SMLM challenge dataset, but lowered Gaussian, B-Spline ($q=3$, $s=2$) and LOG seem to be better within the graphs of Figure 4c.
5. Who well does the algorithm work for arbitrary PSF (e.g. astigmatism, double helix, tetrapod, ...)?
6. The sample is not well described. I suggest adding sketches showing the origami immobilisation strategies and their labels.
7. The wording/language style needs to be improved.
8. The introduction to the single molecule localization pipeline should provide a more detailed explanation of why false signal detection remains a significant challenge. It may be beneficial for the authors to consider integrating relevant sections from Supplementary Information Note 1 into the introduction to address this issue more effectively. This adjustment would

enhance the reader's understanding of the persistence and complexity of false signal detection, thereby emphasizing the need for improved methodologies in the study.

9. The authors' current approach involves utilizing non-maximum suppression (NMS) after image thresholding to detect potential single molecule signals. The authors should clarify why they use both image thresholding and non-maximum suppression (NMS) to detect single molecule signals.

(Remarks on code availability)

The code (Python) was easily runnable and used only default python libraries which are usually installed for scientific purposes (scipy, scikit, numpy & matplotlib). The code allows to reproduce the results provided in the publication. Using the MIT license is a positive choice for ensuring full open-source availability. Better documentation of functions and parameters is recommended to improve code readability and comprehension, as well as to facilitate understanding of important methods described in the publication. Furthermore, I would recommend adding modern python type annotations:

```
def isf_threshold_norm(pfa: float, b: float, w: np.ndarray) -> float:
# isf_threshold_norm: Inverse Survival function(ISF) threshold for a normal distribution
# Parameters:
# pfa = probability of false alarm
# b = background mean
# w = filter kernel
def cfar_segmentation(T: np.ndarray, pfa: float, w: np.ndarray) -> np.ndarray:
# cfar_segmentation: Constant False Alarm Rate (CFAR) segmentation
# Parameters:
# T = filtered image
# pfa = probability of false alarm
# w = filter kernel
```

Reviewer #4

(Remarks to the Author)

In this manuscript, Hekrdla et al present a new molecule detection method for SMLM (and SPT). In current state-of-the-art detection methods, users have to pre-define filter methods to obtain suitable signal/noise ratio for single-molecule detection. Even then, users need to pre-define a signal/noise threshold that identifies molecules. Hekrdla et al propose solutions to both these issues. First of all, they present a mathematical framework to create a theoretically best filter (PMF/MF) which is based on the PSF shape and the signal/noise ratio. Secondly, they perform probabilistic thresholding to allow for a fixed false positive probability during SMLM identification. Both these methods are of interest for the SMLM community, as they address an issue which to this day does not have a satisfactory response (with possibly the exception of DL methods, which have their own downsides).

The manuscript is to the best of my knowledge scientifically solid, clearly written, and has plenty of detail to fully understand the logical steps. Using the experimentally-derived signal/noise ratio (a/b) to obtain a best PMF is especially conceptually interesting.

There are a number of issues that in my opinion should be addressed, however. I want to note that this manuscript in fact has 2 ideas merged in one: a filter-creation method (2.1.1) and probabilistic thresholding (2.1.2/2.1.3), and some of these issues are applicable to only one or the other.

1. On the filter-creation method: Currently, the manuscript is based around fitting in-focus, 'perfect', PSFs, where the authors somewhere in the logic in 3.1.6/3.1.7 further approximate this as a Normal distribution to obtain the PMF shapes in Figure 2. They further only assess their method on 'perfect' PSFs. However, their underlying methodology is more general and would allow for any PSF shape. Effectively, I believe this manuscript would benefit enormously from including more exotic PSF shapes, for instance the Double-Helix PSF used in the SMLM challenge (2013). This would far more clearly show the usability of their filter-creation methodology for arbitrarily shaped PSFs. If this is impossible, this is a limitation of the method, it should be clearly stated as such.

2. On the filter-creation method: A large underlying assumption throughout the manuscript is a Poisson noise environment. This is true for EMCCD cameras, but sCMOS detectors have pixel-dependent noise/gain/offset values (and are getting more popular in recent years). The applicability of the proposed method on sCMOS detectors should be addressed.

3. In the evaluation part of the manuscript (2.2), the authors are always using their probabilistic thresholding method with a variety of filters to assess the performance. I miss a section before this where they compare their probabilistic thresholding method with 'current' methods, which would far more strongly show the power of the probabilistic thresholding part. I would naively envision a plot with x-axis different signal/noise levels, y-axis Jaccard-index (or similar), and lines of e.g. B-spline with fixed values (amongst others), and their probabilistic thresholding method with PMF/MF at varying FP probabilities. This should (naively) show that fixed B-spline would be dependent on signal-noise level, while their method would outperform throughout signal/noise levels.

In addition to this point, I believe the authors should compare their method with deep-learning methods, which are created (partly) with specifically the problem the authors try to solve in mind.

4. In figure 5, the authors showcase their method on experimentally obtained fixed gold nanoparticles. I believe this should be expanded with actual 'blinking' SMLM data as well (STORM/PALM/PAINT). Currently, this method is a good

experimental method to determine a ground-truth (by having high-intensity, low-background gold nanoparticles), and should thus remain included, but having a STORM/PALM/PAINT dataset (and e.g. reporting the number of found localizations and/or show the fully localized images by using a 2D Gaussian fitter) would be of more interest and have much higher statistics than currently the case.

5. The code underlying the manuscript is available, seems to be complete, and could be run without issues, but is organized haphazardly (e.g. often includes test methods, is annotated only sporadically). It would be helpful to create more accessible information, e.g. a Jupyter notebook which is strongly annotated, and *only* does the steps required for the best-possible result (e.g. loading data or simulating data, creating the PMF, and obtaining the localizations with certain FP probability). This would strongly increase the chance it will be widely used by the community.

Smaller points:

6. On the probabilistic thresholding: It is relatively common in the SMLM field to present the Jaccard index (a value that encompasses FP/FN/TP). It might be interesting to also present the data in figure 4 as Jaccard index rather than Probability of Detection to allow for comparison with other manuscripts.

7. On the probabilistic thresholding: The authors choose to find values a/b from using the first 0.25 percentile data taken from the first few frames of raw data. I would strongly advise to change this to randomly selected frames, since in experimental data, especially STORM and PALM, the first few frames normally have a higher localization density than the average data, thus influencing this value finding.

8. In 3.5, the authors mention that the data in figure 5 is created by adding a light source. Why did the authors choose for this rather than decreasing the excitation laser power? This would more closely reflect SMLM datasets (although I do think that the results will be equivalent).

9. While the authors multiple times claim using their method for SMLM and SPT, the unique challenges associated with SPT (especially blurred PSFs) are not addressed, and I'm not sure if they are possible to be addressed (since unknown diffusion leads to unknown blurriness of PSFs, leads to unknown PSF model).

10. In addition to my point 3 above, the users should evaluate and discuss the advantages of their method with existing background-removal pre-processing steps, such as temporal median filtering (Hoogendoorn et al., 2014).

(Remarks on code availability)

See author comments.

Reviewer #5

(Remarks to the Author)

(Remarks on code availability)

The code (Python) was easily executable and used only standard Python libraries that are normally installed for scientific purposes (scipy, scikit, numpy & matplotlib). The code makes it possible to reproduce the results given in the publication. The use of a full open source MIT license is positive. It was difficult for me to customize the code for a SMLM dataset as only long Jupyter notebooks are provided for specific tasks. An additional Jupyter notebook where custom data can be imported and filtered would be desirable.

I would also recommend the author to better document the functions and describe the parameters, otherwise the code and important methods described in the publication are difficult to understand. I would also recommend adding modern Python type annotations:

```
def isf_threshold_norm(pfa: float, b: float, w: np.ndarray) -> float:
# isf_threshold_norm: Inverse Survival function(ISF) threshold for a normal distribution
# Parameters:
# pfa = probability of false alarm
# b = background mean
# w = filter kernel
```

```
def cfar_segmentation(T: np.ndarray, pfa: float, w: np.ndarray) -> np.ndarray:
# cfar_segmentation: Constant False Alarm Rate (CFAR) segmentation
# Parameters:
# T = filtered image
# pfa = probability of false alarm
# w = filter kernel
```

Version 1:

Reviewer comments:

Reviewer #2

(Remarks to the Author)

I was reviewer #2. I wrote "To summarize I think this is a very good paper which should be published and it is a nice contribution to the field."

Now the authors have added even more to the paper and it is more complete.

However, the authors only sent a document with all of the editing marks included. The whole paper is now only red crossed out stuff and blue underlined stuff. I am not going to waste my time trying to decipher this mess.

(Remarks on code availability)

Reviewer #3

(Remarks to the Author)

In the paper titled "Optimized Molecule Detection in Localization Microscopy with Selected False Positive Probability," the authors present a background-adaptive probabilistic thresholding approach. This method allows for precise control over false positive rates in molecule detection, while simultaneously minimizing false negatives, thus enhancing detection accuracy in localization microscopy. The quality of the manuscript has improved significantly after the latest revision, now offering additional valuable insights into the comparison of filtering methods. Overall, I believe the manuscript is now ready for publication; I've included a few suggestions for further refinement.

Concerns:

Inclusion of a Representative SMLM Image of Biological Data: To effectively demonstrate the capabilities of the developed algorithm, including a representative SMLM image of biological data (such as data from the SMLM challenge without ground truth) would be ideal. This should illustrate a comparison of biological data at varying PFP levels, with renderings of the SMLM image (potentially with a zoomed-in view) and a sample frame highlighting detections, similar to Figure 3. This addition would provide a concise visual summary of the detailed descriptions in the text.

The generalization for 3D data is a valuable enhancement, but to make the algorithm fully applicable across SMLM applications, a code example or further explanation for the z-parameterized PSF is recommended.

Authors claim: "... probabilistic thresholding has a linear or constant complexity that is computationally efficient, making it computationally efficient for real-time applications. ..." - do you have hints on time resolution from simulations – number would be interesting? – page 13

Paragraph 2.1.1. Optimal Filtering optimal regarding what? Rephrase

Page 9 – eq. 7-9 the parameters a and b are explained on page 13 for the first time.

Suppl.3/3.3. – "Using the Gaussian CCDF puts the derivation in a broader context, and may be more convenient in implementation." – in which context, specify.

Suppl.3/3.1. – "The resulting PDF is for reasonably high background well approximated by Gaussian PDF as ..." – what is a 'reasonably high background'? Can you provide some boundaries?

Minor:

Reference: A. Neubeck et al., Efficient Non-Maximum Suppression, 2006 applied NMS directly to the filtered image. The probabilistic threshold calculated here could be integrated directly into the NMS algorithm, potentially eliminating the need for a separate image thresholding step.

(Remarks on code availability)

The code is fine - did not change significantly since last revision

Reviewer #4

(Remarks to the Author)

The authors fully and satisfactorily responded to all my remarks, and these are all fully addressed in the revised manuscript. I very much appreciate the extension to 3D and their discussion on the sCMOS noise model. I understand the rationale of the authors to not implement some of my suggestions. The added exemplary Python code works wonderfully.

I congratulate the authors on their work and wish them all the best on the continuation of this exciting research line.

(Remarks on code availability)

The added exemplary Python code works wonderfully.

Reviewer #5

(Remarks to the Author)

(Remarks on code availability)

The code (Python) was easily runnable and used only default python libraries which are usually installed for scientific purposes (scipy, scikit, numpy & matplotlib). The code allows to reproduce the results provided in the publication. A code example for the z-parameterized PSF is missing.

Manuscript NCOMMS-24-14455A

Here is a point-by-point response to the concerns raised by the reviewers:

We would like to thank all reviewers for their time and valuable and constructive feedback. The remarks were thought-provoking and raised important points that enabled us to improve the manuscript.

Reviewer #2 (Remarks to the Author):

This paper shows a new method for detection of single molecules in single molecule localization microscopy. The author's method (Poisson Matched Filter) incorporates a filter template with Poisson statistics, resulting in a better-performing method than previously available. The paper is well put together and is very thorough.

The authors should add the following reference, which shows a similar Monte-Carlo approach for performance evaluation of detection methods:

Křížek, et al, "Minimizing detection errors in single molecule localization microscopy," Opt. Express 19, 3226-3235 (2011).

We included the reference and the context.

Is it possible to evaluate more of the single molecule challenge data? Some of the data sets present in the challenge (2013 and 2016 data) have higher levels of noise, or higher densities of molecules.

We evaluated more of the single molecule challenge data and included the results in Supporting Information (Supplementary Figure 7).

In future work, it would be nice to see further imaging of DNA origami structures and also of structures within cells. This should show the authors method effectively reduces false molecule detection in real data.

We thank the reviewer for the suggestion. Indeed, we plan to incorporate the detection method within selected localization SW to enable imaging of DNA origami structures and cellular structures.

To summarize I think this is a very good paper which should be published and it is a nice contribution to the field.

Reviewer #3 (Remarks to the Author):

Concerns:

1. The method is well described mathematically, but its advantages over other methods are not sufficiently demonstrated. The graphics lack a comprehensive explanation, particularly regarding the quantification of comparative results in relation to other methods.

We added a more thorough explanation as suggested.

2. Introduction and abstract have to put the methodology into a broader context and emphasize on the importance of this result for the SMLM field. Additionally, it is important to highlight the advantages of the described method in comparison to new AI-based methods (e.g., BGnet, a deep neural network).

We added broader context in the abstract and introduction as suggested. We also added discussion highlighting the advantages of our approach in comparison to new AI-based methods (section Supplementary Note 9, comparison of the speed is added to the main text). The primary advantage of our approach compared to the Deep Learning (DL) approaches is its predictability and interpretability as it is based on classical analytical methods. The lack of interpretability is one of the biggest issues for DL models that limits its use in high-stake applications where understanding the decision-making process is crucial.

3. A computation speed comparison of the algorithms described is missing. Is this introduced algorithm suitable for real time applications which suffer from false signal detection?

The speed is thoroughly discussed in Supplementary note 4, where we analyze the asymptotic computational complexity (see the references (R.C. Gonzalez, R.E. Woods, Digital image processing, Pearson, 2018), (I.T. Young, et al., Fundamentals of image processing, Delft University of Technology, 1998)) of different possible implementations of the proposed filtering and background estimation. In all cases, the computational complexity is no worse than linear, allowing for an inexpensive real-time implementation.

4. As of Figure 4 the author mentioned that PMF & MF are the best filters for the SMLM challenge dataset, but lowered Gaussian, B-Spline ($q=3$, $s=2$) and LOG seem to be better within the graphs of Figure 4c.

According to the ROC characteristics, PMF and MF are the best performers even in the graphs of Figure 4c according to standards in evaluation of ROC curves (T. Fawcett, An introduction to ROC analysis, Pattern Recognition Letters 27, 861-874, 2006). As pointed out, LoG, B-Spline ($q=3$, $s=2$) perform as good as PMF and MF for higher Probability of False Positive (10^{-3} , 10^{-2}).

5. How well does the algorithm work for arbitrary PSF (e.g. astigmatism, double helix, tetrapod, ...)?

In principle, the algorithm is general and works for any model of PSF. We added the solutions for three general approaches (multiplane, z-parametrized PSF - covering astigmatism, double helix, tetrapod, and z-projection) to the text. To demonstrate the detection, we extended the algorithm to 3D and implemented the detection for multiplane imaging and evaluated its performance on simulated data (Supporting Information Note 11). The complete extension of the method to other shapes of PSF and rigorous evaluation of performance will be the focus of our future studies. We see

great potential of our method in this aspect and thank the reviewer for this stimulating comment!

6. The sample is not well described. I suggest adding sketches showing the origami immobilisation strategies and their labels.

We added the sketch and explanation to describe the sample better, we thank the reviewer for pointing it out!

7. The wording/language style needs to be improved.

We modified the wording and language.

8. The introduction to the single molecule localization pipeline should provide a more detailed explanation of why false signal detection remains a significant challenge. It may be beneficial for the authors to consider integrating relevant sections from Supplementary Information Note 1 into the introduction to address this issue more effectively. This adjustment would enhance the reader's understanding of the persistence and complexity of false signal detection, thereby emphasizing the need for improved methodologies in the study.

As suggested, we added more information to the introduction and compiled some of the text in Supplementary Note 1 to provide context and highlight the benefits of our method.

9. The authors' current approach involves utilizing non-maximum suppression (NMS) after image thresholding to detect potential single molecule signals. The authors should clarify why they use both image thresholding and non-maximum suppression (NMS) to detect single molecule signals.

The proposed molecule detection procedure is derived based on the assumption of single molecule detection. The resulting detector searches for the maximum of the test statistic, which it compares with the detection threshold (Equation 26 in the manuscript). If we apply this procedure to real data with multiple molecules, we have to adapt the detection algorithm so that instead of a global maximum, we search for a local maximum of approximately the size of one molecule. A discussion of the generalization beyond this condition (denoted as a crowded field problem) is given in Supplementary Note 5.

Reviewer #3 (Remarks on code availability):

The code (Python) was easily runnable and used only default python libraries which are usually installed for scientific purposes (scipy, scikit, numpy & matplotlib). The code allows to reproduce the results provided in the publication. Using the MIT license is a positive choice for ensuring full open-source availability. Better documentation of functions and parameters is recommended to improve code readability and comprehension, as well as to facilitate understanding of important methods described in the publication. Furthermore, I would recommend adding modern python type annotations:

```

def isf_threshold_norm(pfa: float, b: float, w: np.ndarray) -> float:
# isf_threshold_norm: Inverse Survival function(ISF) threshold for a normal distribution
# Parameters:
# pfa = probability of false alarm
# b = background mean
# w = filter kernel
def cfar_segmentation(T: np.ndarray, pfa: float, w: np.ndarray) -> np.ndarray:
# cfar_segmentation: Constant False Alarm Rate (CFAR) segmentation
# Parameters:
# T = filtered image
# pfa = probability of false alarm
# w = filter kernel

```

We extended documentation and modified annotations based on the suggestions.

Reviewer #4 (Remarks to the Author):

In this manuscript, Hekrdla et al present a new molecule detection method for SMLM (and SPT). In current state-of-the-art detection methods, users have to pre-define filter methods to obtain suitable signal/noise ratio for single-molecule detection. Even then, users need to pre-define a signal/noise threshold that identifies molecules. Hekrdla et al propose solutions to both these issues. First of all, they present a mathematical framework to create a theoretically best filter (PMF/MF) which is based on the PSF shape and the signal/noise ratio. Secondly, they perform probabilistic thresholding to allow for a fixed false positive probability during SMLM identification. Both these methods are of interest for the SMLM community, as they address an issue which to this day does not have a satisfactory response (with possibly the exception of DL methods, which have their own downsides).

The manuscript is to the best of my knowledge scientifically solid, clearly written, and has plenty of detail to fully understand the logical steps. Using the experimentally-derived signal/noise ratio (a/b) to obtain a best PMF is especially conceptually interesting.

There are a number of issues that in my opinion should be addressed, however. I want to note that this manuscript in fact has 2 ideas merged in one: a filter-creation method (2.1.1) and probabilistic thresholding (2.1.2/2.1.3), and some of these issues are applicable to only one or the other.

1. On the filter-creation method: Currently, the manuscript is based around fitting in-focus, 'perfect', PSFs, where the authors somewhere in the logic in 3.1.6/3.1.7 further approximate this as a Normal distribution to obtain the PMF shapes in Figure 2. They further only assess their method on 'perfect' PSFs. However, their underlying methodology is more general and would allow for any PSF shape. Effectively, I believe this manuscript would benefit enormously from including more exotic PSF shapes, for instance the Double-Helix PSF used in the SMLM challenge (2013). This would far more clearly show the usability of their filter-creation methodology for arbitrarily shaped PSFs. If this is impossible, this is a limitation of the method, it should be clearly stated as such.

We thank the reviewer for this suggestion. We agree that the methodology is more general and to demonstrate it, we extended the method to 3D and implemented the detection for multiplane imaging, as an example and a significant extension of the method. We added a solution to the manuscript that generalizes the method to 3D using three different approaches (multiplane, z-parametrized PSF - covering astigmatism, double helix, tetrapod, and z-projection) to the text. To demonstrate the detection, we extended the algorithm to 3D and implemented the detection for multiplane imaging and evaluated its performance on simulated data (Supporting Information Note 11). The complete extension of the method to other shapes of PSF and rigorous evaluation of performance will be the focus of our future studies.

2. On the filter-creation method: A large underlying assumption throughout the manuscript is a Poisson noise environment. This is true for EMCCD cameras, but sCMOS detectors have pixel-dependent noise/gain/offset values (and are getting more popular in recent years). The applicability of the proposed method on sCMOS detectors should be addressed.

We extended the work and addressed this issue. It is comprehensively described in Supplementary Note 8. In addition to this, we extended the discussion of the noise in EMCCD cameras for the cases, when the Poisson noise does not apply, added to Supplementary Note 10. We thank the reviewer for the stimulating comment!

3. In the evaluation part of the manuscript (2.2), the authors are always using their probabilistic thresholding method with a variety of filters to assess the performance. I miss a section before this where they compare their probabilistic thresholding method with 'current' methods, which would far more strongly show the power of the probabilistic thresholding part. I would naively envision a plot with x-axis different signal/noise levels, y-axis Jaccard-index (or similar), and lines of e.g. B-spline with fixed values (amongst others), and their probabilistic thresholding method with PMF/MF at varying FP probabilities. This should (naively) show that fixed B-spline would be dependent on signal-noise level, while their method would outperform throughout signal/noise levels. In addition to this point, I believe the authors should compare their method with deep-learning methods, which are created (partly) with specifically the problem the authors try to solve in mind.

We appreciate the idea to include a comparison of the probabilistic thresholding method with current methods across different signal-to-noise levels, however we would like to stay with our current approach, as it provides more comprehensive understanding of the detection problem, while demonstrating the advantage and implementation of the probabilistic thresholding to assess the filtering methods. We tried to highlight the benefits of the probabilistic thresholding compared to fixed threshold in the text and to phrase it better.

We added comparison to deep-learning methods to Supporting Information Note 9 and stressed the advantages of our approach and those of DL.

4. In figure 5, the authors showcase their method on experimentally obtained fixed gold nanoparticles. I believe this should be expanded with actual 'blinking' SMLM data as well (STORM/PALM/PAINT). Currently, this method is a good experimental method to determine a ground-truth (by having high-intensity, low-background gold nanoparticles), and should thus remain included, but having a STORM/PALM/PAINT dataset (and e.g. reporting the number of found localizations and/or show the fully localized images by using a 2D Gaussian fitter) would be of more interest and have much higher statistics than currently the case.

We plan to integrate our detection method to the localization SW in our future work. The current work focuses specifically on the detection step. We agree that actual blinking data would provide higher statistics and could be of interest to the readers, but after considering the benefits and drawbacks, we decided to keep the current evaluation. As you noted, our approach using high-intensity, low-background nanoparticles is excellent to obtain ground truth that enables to evaluate the detection performance. While blinking data provide higher statistics, they lack a reliable ground truth, making it impossible to accurately assess detection accuracy, especially false negatives. We could provide the total number of found localizations, however without accurate information on how many of those are false positives and what are missed - false negative - detection, this number can be misleading and difficult to interpret - both for us as well as for the readers. This is in agreement with (Sage, Daniel, et al. "Quantitative evaluation of software packages for single-molecule localization microscopy." Nature methods 12.8 (2015): 717-724.), (Sage, Daniel, et al. "Super-resolution fight club: assessment of 2D and 3D single-molecule localization microscopy software." Nature methods 16.5 (2019): 387-395.) and context of the references.

5. The code underlying the manuscript is available, seems to be complete, and could be run without issues, but is organized haphazardly (e.g. often includes test methods, is annotated only sporadically). It would be helpful to create more accessible information, e.g. a Jupyter notebook which is strongly annotated, and *only* does the steps required for the best-possible result (e.g. loading data or simulating data, creating the PMF, and obtaining the localizations with certain FP probability). This would strongly increase the chance it will be widely used by the community.

We cleared the code as suggested! We created an example notebook that performs the detection on simulated data by the method we developed. Additionally, we separated the testing part, the part of the simulations, SMLM challenge and experimental data to make it more useful for the users and easier to orient. These parts enable full reproduction of the results presented in our paper.

Smaller points:

6. On the probabilistic thresholding: It is relatively common in the SMLM field to present the Jaccard index (a value that encompasses FP/FN/TP). It might be interesting to also present the data in figure 4 as Jaccard index rather than Probability of Detection to allow for comparison with other manuscripts.

We have added a comparison using the Jaccard index to the Supporting Information (Supplementary Figure 8) to enable direct comparison with other manuscripts, as

suggested. However, we maintain that ROC analysis is more appropriate for our main results due to the nature of molecule detection in SMLM. Unlike image segmentation, where false positives (FP) and false negatives (FN) may have equal importance, in molecule detection, FPs have a more significant negative impact. They can create non-existent structures in the super-resolved image and propagate errors through subsequent SMLM processing steps. Our method allows for control of the FP rate while minimizing FN, making ROC evaluation particularly suitable. The Probability of Detection metric in Figure 4 effectively captures this trade-off. We have expanded our discussion in the manuscript to clarify the rationale behind our choice of metrics, referencing recent community guidelines on metric selection in image processing tasks (Hein-Reinke et al., 2024).

7. On the probabilistic thresholding: The authors choose to find values a/b from using the first 0.25 percentile data taken from the first few frames of raw data. I would strongly advise to change this to randomly selected frames, since in experimental data, especially STORM and PALM, the first few frames normally have a higher localization density than the average data, thus influencing this value finding.

We modified this as suggested.

8. In 3.5, the authors mention that the data in figure 5 is created by adding a light source. Why did the authors choose for this rather than decreasing the excitation laser power? This would more closely reflect SMLM datasets (although I do think that the results will be equivalent).

For the purposes of this experiment, we aimed to control the laser excitation and the noise level as independently as possible. Our main motivation behind utilizing another light source was to avoid modifying the parameters of the laser excitation (beam path, beam shape, etc) as a side-effect of adding artificial noise, which we believe would be more likely when changing the excitation laser power. That being said, we agree that decreasing the excitation laser power instead would yield equivalent results.

9. While the authors multiple times claim using their method for SMLM and SPT, the unique challenges associated with SPT (especially blurred PSFs) are not addressed, and I'm not sure if they are possible to be addressed (since unknown diffusion leads to unknown blurriness of PSFs, leads to unknown PSF model).

Addressing the challenge of blurred PSFs in SPT would be related to the generalization of the detection method and could be solved for example by using a larger PSF sigma in the model. However, we have not tested it specifically for the case and therefore agree that we should not generalize for this case. We deleted mentions of applications for SPT and hope to be able to address this issue in the future.

10. In addition to my point 3 above, the users should evaluate and discuss the advantages of their method with existing background-removal pre-processing steps, such as temporal median filtering (Hoogendoorn et al., 2014).

We agree that to support our claim (background removal is not theoretically justified when background adaptive approach is used) we should provide complete comparison with existing background-removal methods as suggested. We modified the text to better represent our case. We already compared our approach to selected methods that contain background removing steps (DoG, DoA, LG, B-Spline). It is an interesting point for future studies to also include other methods such as temporal medial filtering.

Reviewer #4 (Remarks on code availability):

See author comments.

Reviewer #5 (Remarks to the Author):

Reviewer #5 (Remarks on code availability):

The code (Python) was easily executable and used only standard Python libraries that are normally installed for scientific purposes (scipy, scikit, numpy & matplotlib). The code makes it possible to reproduce the results given in the publication. The use of a full open source MIT license is positive. It was difficult for me to customize the code for a SMLM dataset as only long Jupyter notebooks are provided for specific tasks. An additional Jupyter notebook where custom data can be imported and filtered would be desirable.

I would also recommend the author to better document the functions and describe the parameters, otherwise the code and important methods described in the publication are difficult to understand. I would also recommend adding modern Python type annotations:

```
def isf_threshold_norm(pfa: float, b: float, w: np.ndarray) -> float:
# isf_threshold_norm: Inverse Survival function(ISF) threshold for a normal distribution
# Parameters:
# pfa = probability of false alarm
# b = background mean
# w = filter kernel
```

```
def cfar_segmentation(T: np.ndarray, pfa: float, w: np.ndarray) -> np.ndarray:
# cfar_segmentation: Constant False Alarm Rate (CFAR) segmentation
# Parameters:
# T = filtered image
# pfa = probability of false alarm
# w = filter kernel
```

We thank you for running and testing the code and for the suggestions! We made modifications accordingly. We added documentation and changed annotations based

on your suggestions. We additionally cleared the code and separated files for testing, simulations, SMLM challenge and experimental data to make the codes more useful.

Overall, we would like to express gratitude to all reviewers for their time, expertise and valuable feedback. We found the review process exceptionally constructive with intriguing and thought-provoking comments and fair criticism. We believe to have improved our manuscript and made it more impactful.

Here is a point-by-point response to the concerns raised by the reviewers:

We would like to thank all reviewers for valuable feedback. Here is a response to the raised concerns.

Reviewer #3 (Remarks to the Author):

Concerns:

Inclusion of a Representative SMLM Image of Biological Data: To effectively demonstrate the capabilities of the developed algorithm, including a representative SMLM image of biological data (such as data from the SMLM challenge without ground truth) would be ideal. This should illustrate a comparison of biological data at varying PFP levels, with renderings of the SMLM image (potentially with a zoomed-in view) and a sample frame highlighting detections, similar to Figure 3. This addition would provide a concise visual summary of the detailed descriptions in the text.

We agree that the capabilities would be effectively demonstrated on biological data and as part of the SMLM software. As suggested in the previous round of review, we already added demonstration and evaluation of performance on sets from the SMLM challenge data (possible to do only on data with known ground truth). We believe that this information, together with information from experimental data in Figure 5 and simulated data in Figure 3, provide a comprehensive quantitative evaluation about the capabilities of the developed detection method. Additional demonstration of the detection on SMLM challenge data without ground truth might bring a qualitative and visually appealing demonstration of capabilities (similar to Figure 3 or Figure 5), but does not add any quantitative information.

To demonstrate the final impact on the rendered image, an implementation within the SMLM software is necessary, which is a task we plan to do within the future work.

The generalization for 3D data is a valuable enhancement, but to make the algorithm fully applicable across SMLM applications, a code example or further explanation for the z-parameterized PSF is recommended.

We rephrased the paragraph about the z-parametrized PSF to make this more clear. We thank for the suggestion.

Authors claim: “... probabilistic thresholding has a linear or constant complexity that is computationally efficient, making it computationally efficient for real-time applications. ...”
- do you have hints on time resolution from simulations – number would be interesting? –
page 13

We added this information to the Supplementary Note 10. Thank you for the suggestion!

Paragraph 2.1.1. Optimal Filtering optimal regarding what? Rephrase

We rephrased it.

Page 9 – eq. 7-9 the parameters a and b are explained on page 13 for the first time.

It is defined also above, on page 4 when the a/b ratio is defined.

Suppl.3/ 3.3. – “Using the Gaussian CCDF puts the derivation in a broader context, and may be more convenient in implementation.” – in which context, specify.

We deleted this statement, it was misleading.

Suppl.3/ 3.1. – “The resulting PDF is for reasonably high background well approximated by Gaussian PDF as ...” – what is a ‘reasonably high background’? Can you provide some boundaries?

It is for 5 photons and above, we added this and included a relevant reference.

Minor:

Reference: A. Neubeck et al., Efficient Non-Maximum Suppression, 2006 applied NMS directly to the filtered image. The probabilistic threshold calculated here could be integrated directly into the NMS algorithm, potentially eliminating the need for a separate image thresholding step.

We added the reference

Reviewer #5 (Remarks on code availability):

The code (Python) was easily runnable and used only default python libraries which are usually installed for scientific purposes (scipy, scikit, numpy & matplotlib). The code allows to reproduce the results provided in the publication. A code example for the z-parameterized PSF is missing.

We believe that the reviewer means the code for the multiplane detection as an example for 3D detection. We did not include an example code for this particular case as it is only a minor part of the work and a demonstration of principle. The complete code to reproduce the multiplane experiments is included.